# M72 Fusion Proteins in Nanocapsules Enhance BCG Efficacy Against Bovine Tuberculosis in a Mouse Model

**DOI:** 10.3390/pathogens14060592

**Published:** 2025-06-16

**Authors:** Federico Carlos Blanco, Renée Onnainty, María Rocío Marini, Laura Inés Klepp, Elizabeth Andrea García, Cristina Lourdes Vazquez, Ana Canal, Gladys Granero, Fabiana Bigi

**Affiliations:** 1Instituto de Agrobiotecnología y Biología Molecular, (IABIMO) INTA-CONICET, Instituto de Biotecnología, CICVyA, Instituto Nacional de Tecnología Agropecuaria (INTA), N. Repetto and De los Reseros, Hurlingham 1686, Argentina; blanco.federico@inta.gob.ar (F.C.B.); klepp.laura@inta.gob.ar (L.I.K.); garcia.elizbeth@inta.gob.ar (E.A.G.);; 2Unidad de Investigaciones y Desarrollo en Tecnología Farmacéutica (UNITEFA)-CONICET, Departamento de Ciencias Farmacéuticas, Facultad de Ciencias Químicas, Universidad Nacional de Córdoba, Córdoba 5000, Argentina; ronnainty@unc.edu.ar; 3Laboratorio de Anatomía Patológica, Facultad de Ciencias Veterinarias, Universidad Nacional del Litoral, Santa Fe 2000, Argentina

**Keywords:** *Mycobacterium bovis*, bovine tuberculosis, mice, vaccine, chitosan nanocapsules, M72

## Abstract

*Mycobacterium bovis* is the causative pathogen of bovine tuberculosis (bTB), a disease that affects cattle and other mammals, including humans. Currently, there is no efficient vaccine against bTB, underscoring the need for novel immunization strategies. The M72 fusion protein, composed of three polypeptides derived from *Mycobacterium tuberculosis* and *M. bovis*, has demonstrated protective efficacy against *M. tuberculosis* in clinical trials when combined with the AS01E adjuvant. Given the established efficacy of nanocapsule formulations as vaccine delivery systems, this study evaluated a novel immunization strategy combining BCG with either full-length M72 or a truncated M72 fused to a streptococcal albumin-binding domain (ABDsM72). Both antigens were encapsulated in chitosan/alginate nanocapsules and assessed in a murine *M. bovis* challenge model. Priming with BCG followed by an M72 boost significantly improved splenic protection compared to BCG alone, but it did not enhance pulmonary protection. Notably, boosting with ABDsM72 further increased the proportion of CD4+KLRG1-CXCR3+ T cells in the lungs of *M. bovis*-challenged mice, a key correlate of protective immunity. These findings demonstrate that chitosan/alginate-encapsulated antigens enhance BCG-induced immunity, supporting their potential as next-generation vaccine candidates for bTB control.

## 1. Introduction

*Mycobacterium bovis*, a member of the *Mycobacterium tuberculosis* complex (MTBC), is the causative agent of bovine tuberculosis (bTB), a chronic respiratory disease affecting a wide range of mammals, including humans. The persistent nature of bTB contributes to its epidemiological complexity [1]. In addition to cattle, bTB infects other livestock species such as sheep, goats, and pigs, causing detrimental effects on animal health and agricultural productivity [2].

As a zoonotic disease, bTB primarily poses a risk to rural workers in close contact with livestock. Additionally, consumption of contaminated unpasteurized dairy products represents another transmission route [3]. The economic consequences of bTB are substantial, including direct losses (e.g., reduced milk productivity and early culling of animals) and indirect losses (arising from trade restrictions on livestock products). In South America and the Caribbean, the cattle population is estimated to be 374 million, with 70% located in high-prevalence zones and 17% in areas approaching disease-free status. Brazil, with 169 million cattle, reported a bTB prevalence of 3–6%, while Paraguay and Uruguay, with 9 million and 11.7 million cattle, respectively, exhibit markedly lower rates (0.25% and 0.01%) [4]. In Argentina, bTB is a notifiable disease, and animals that react positively to the official test are required to be slaughtered https://www.argentina.gob.ar/normativa/nacional/resoluci%C3%B3n-128-2012-195314 (accessed on 10 April 2025).

Vaccination remains the most effective strategy for infectious disease control, yet no approved vaccine exists for bTB. Most experimental trials have utilized the human Bacille Calmette–Guérin (BCG) TB vaccine. However, BCG vaccination interferes with the tuberculin skin test, the primary diagnostic tool for bTB, which relies on the purified protein derivative of *M. bovis* (PPDb) [5,6,7].

Recent research has explored multiple vaccination strategies against bTB, which have been tested in both animal models and natural hosts of *M. bovis* [5,6,7]. The key limitation of bTB vaccines is their interference with conventional diagnostics. However, advancements in DIVA (Differentiating Infected from Vaccinated Animals) diagnostics, particularly supplementary assays, now allow for the discrimination between BCG-vaccinated and naturally infected animals. In this regard, the antigens ESAT-6, CFP-10, and Rv3615c—used as a cocktail or as a fused protein—have demonstrated a diagnostic performance comparable to PPDb in intradermal tuberculin tests [8]. They offer similar sensitivity while improving specificity by avoiding cross-reactivity with non-tuberculous mycobacteria or BCG vaccination, as these antigens are absent from BCG strains [9].

While subunit TB vaccines (e.g., the promising M72/AS01E) are advancing in human clinical trials [10], only a limited number of protein- or DNA-based bTB candidates have been evaluated preclinically, with even fewer progressing to field trials. Subunit vaccines offer DIVA compatibility due to their reliance on multiple antigens, although their protective efficacy is expected to be modest. As such, they may be most effective as boosters following BCG priming or in combination with attenuated *M. bovis* strains, ideally paired with a DIVA-compliant diagnostic.

An emerging immunological strategy, which was initially explored in cancer therapy, is mucosal immunization via albumin hitchhiking. Albumin, the most abundant serum protein, traverses epithelial barriers by binding neonatal Fc receptors (FcRn). This approach leverages endogenous albumin as a carrier for vaccine antigens. Antigen–albumin conjugation can occur either through lipid tagging (exploiting albumin’s fatty acid transport function) or by genetic fusion with the albumin-binding domain (ABD) of streptococcal protein G [11,12]. The latter method avoids chemical modification, preserving antigen integrity. Because FcRn is widely expressed in epithelial and endothelial tissues, ABD-fused antigens exhibit enhanced uptake into the lymphatic system, promoting efficient immune priming [13]. For optimal immunity, both antigens and adjuvants must target lymph node-resident immune cells. In this regard, chitosan nanoparticles have been widely employed as effective nanodelivery systems in vaccine development [14,15]. While chitosan-based vaccines are primarily employed in mucosal vaccination due to their mucoadhesive properties [15], they have also demonstrated efficacy when administered via non-mucosal routes. For instance, chitosan nanoparticles encapsulating the SARS-CoV-2 spike glycoprotein elicited immune responses in the respiratory tract following intraperitoneal inoculation [16]. This finding suggests their capacity for systemic dissemination through lymphatic trafficking [17].

In this study, we combined the attributes of chitosan nanoparticles with the immunogenic properties of M72 antigens, along with ABD-mediated antigen delivery, in a BCG-based prime-boost vaccine strategy against bTB in a mouse model. Although mice do not fully replicate the disease course of tuberculosis that occurs in natural hosts, they remain the most widely used animal model for preclinical testing of TB vaccines due to their manageable size, the availability of well-characterized immune tools for mice, and the diversity of genetically modified strains.

## 2. Materials and Methods

Crystalline vitamin D3 (Vit D3) was generously donated by the Bagó S.A. laboratory (Buenos Aires, Argentina). Corn oil was derived from commercial cooking oil (Cocinero, Argentina). Polyethylene glycol sorbitan monooleate, Tween^®^ 80, low-molecular-weight chitosan (CS; MW: ~50,000–190,000 Da; degree of deacetylation: ~85%) and low-viscosity sodium alginate (ALG; MW: ~4500 Da; viscosity: ~8 cP; α-L-guluronate content: 39%; β-D-mannuronate content: 61%) were provided by Sigma Aldrich (Buenos Aires, Argentina).

### 2.1. Cloning and Expression of Recombinant Proteins in Escherichia coli

The *M72* gene was assembled by sequentially cloning three PCR-amplified fragments—the C-terminal fragment of *pepA* (*Rv0125*; residues 192–323), full-length *ppe18* (*Rv1196*), and the N-terminal fragment of *pepA* (residues 1–195)—into the pGEM-T Easy vector (Promega (Madison, WI, USA)) using *M. bovis* AF2122/97 genomic DNA as a template and the primers listed in Appendix A. The complete M72 sequence was excised with either EcoRI/HindIII or EcoRI/KpnI and subcloned into pRSET-B (Invitrogen (Waltham, MA, USA)) to generate pRSET-B-M72 or into an ABD domain-containing plasmid to produce pMG-AMP-ABDsM72. The *ABDM72* fragment was then digested with XbaI and inserted into the NheI site of pRSET-A, resulting in pRSET-A-ABDsM72.

The plasmids pRSET-B-M72 and pRSET-A-ABDM72 were used to transform *E. coli* BL21(DE3)pLysS and the expression of the recombinant proteins was induced following the standard protocols described elsewhere https://assets.thermofisher.com/TFS-Assets/LSG/manuals/prset_man.pdf (accessed on 10 April 2025). The recombinant proteins were purified by affinity chromatography using nickel-charged resin.

A 50 mL overnight saturated culture of *E. coli* BL21 (DE3) harbouring pRSET recombinant plasmids was inoculated into 500 mL of LB medium supplemented with 50 μg/mL ampicillin. The cells were grown at 37 °C with shaking (220 rpm) until the optical density at 600 nm (OD600) reached 0.3. Protein expression was induced by adding 1 mM isopropyl-β-D-thiogalactoside (IPTG), followed by incubation for 3 h.

The cells were harvested by centrifugation (4000× *g*, 15 min, 4 °C) and resuspended in 20 mL of lysis buffer (100 mM NaH_2_PO_4_, 10 mM Tris-Cl, 8 M urea, pH 8.0). Cell disruption was performed using a Precellys homogenizer (Bertin Technologies, Montigny-le-Bretonneux, France), followed by DNase I treatment (10 μg/mL, 15 min on ice). The lysate was centrifuged (12,000× *g*, 30 min, 4 °C) to remove the insoluble debris.

The clarified supernatant was incubated with 0.5 mL of Ni Sepharose High-Performance resin (GE Healthcare, Chicago, IL, USA) for 2 h at 4 °C with gentle rotation. The resin was pelleted (1000 g, 3 min), washed five times with 50 mL of wash buffer (100 mM NaH_2_PO_4_, 10 mM Tris-Cl, 8 M urea, 20 mM imidazole, pH 6.3), and then transferred to a polypropylene column. After an additional wash, bound proteins were eluted using a step gradient of imidazole (250 mM to 1 M) in elution buffer (100 mM NaH_2_PO_4_, 10 mM Tris-Cl, 8 M urea, pH 8.0).

The eluted fractions’ protein concentration was measured using the Micro BCA Protein Assay Kit (Pierce, Thermo Fisher Scientific, Waltham, MA, USA). The purified proteins were aliquoted and stored at −80 °C until further use.

The protein purification procedure was carefully performed to minimize contamination with LPS or endotoxins.

### 2.2. Antigen–Chitosan/Alginate Nanocapsule (NC) Carrier Preparation

Following our previous protocol (Onnainty et al., 2024) [18], nanocapsules (NCs) were prepared using the solvent displacement technique. Briefly, CS NCs were synthesized by dissolving 0.133 mg of vit D3 (Sigma-Aldrich cat: C9756, St. Louis, MO, USA) in 70 µL of corn oil, 2500 µL of acetone, and 10 µL of the surfactant Tween^®^80. The organic phase was immediately transferred to 5000 µL of an aqueous 0.1% *w*/*v* CS solution and stirred vigorously for 30 min. Finally, the acetone was removed with magnetic stirring overnight under a fume hood and protected from light. The volume was adjusted to 5 mL with Milli-Q water (Milli-Q Purification System, Millipore Bedford, USA). The formulation was then ultracentrifuged at 15,000 rpm (21,000× *g*) for 1 h at 15 °C. For antigen binding (M72 or ABDsM72), 750 μL of an aqueous suspension of CS NPs (1 mg/mL) was incubated with 67 μL or 98 µL of antigen for 20 min at room temperature with stirring. Subsequently, CS/ALG-loaded antigen NPs were obtained by incubating CS NPs loaded with antigen with 200 µL of an aqueous solution of ALG (6.7 mg/mL) under magnetic stirring, and the mixture was stirred for 30 min at room temperature.

The non-loaded CS/ALG NCs’ hydrodynamic radius, measured by dynamic light scattering (DLS) (Zetasizer Nano ZS, Malvern, UK), was 341 nm, and the Z-potential was −47.6 mV.

The hydrodynamic radius of the loaded CS/ALG NCs, measured by dynamic light scattering (DLS) using a Zetasizer Nano ZS (Malvern, UK), showed a predominant population below 200 nm, with a Z-potential of −38 mV (Appendix A).

After freeze-drying, the CS/ALG NCs, CS/ALG-loaded M72 NCs, and CS/ALG-loaded ABDsM72 NCs showed a rounded structure consisting of a dense oily core surrounded by a light layer of entangled polymers in scanning electron microscopy (SEM) images (Appendix A), which were captured using an FE-SEM Σigma analytical scanning electron microscope at an intensity of 5 kVon (Carl Zeiss Sigma, LAMARX laboratory, FAMAF, UNC). After resuspension of the solid sample in water, the hydrodynamic radius of the NCs were predominantly under 200 nm. The Z-potential values were −25 mV for the CS/ALG NCs loaded with M72, −28 mV for those loaded with ABDsM72, and −38 mV for non-loaded CS/ALG NCs. These results were consistent with the SEM measurements, further confirming the accuracy of the initial findings (Appendix A). Successful antigen loading into the NCs was confirmed by SDS-PAGE analysis (Appendix A). The amounts of the recombinant proteins per microgram of NCs were 0.0018–0.0011 μg and 0.0044–0.0026 μg for M72 and ABDsM72, respectively. The vaccines were formulated using a single batch of NCs to ensure consistency across all preparations. The freeze-dried CS/ALG-loaded M72 NCs are referred to as the M72 vaccine and freeze-dried CS/ALG-loaded ABDsM72 NCs are referred to as the ABDsM72 vaccine.

### 2.3. Bacterial Strains, Media, and Growth Conditions

For all experiments, the *M. bovis* strains were grown in supplemented Middlebrook media (7H9/7H10) containing albumin–dextrose–pyruvate (ADP) and Tween 80 (0.05%) at 37 °C. Fresh cultures of BCG and *M. bovis* NCTC10772 (the challenge strain) were processed as previously described [19] to prevent clumping prior to use.

### 2.4. Mouse Vaccination and Infection

Eight female BALB/c mice (6 weeks old) per group were subcutaneously immunized with BCG Pasteur plus either M72 or ABDsM72 formulated in chitosan nanoparticles, which was administered at separate sites (neck and tail base). Booster immunizations with the respective nanovaccines were given at 2 and 4 weeks post-primary vaccination. The control groups received either PBS or BCG alone. The experimental groups were as follows: (i) BCG alone (1 × 10^5^ CFUs), (ii) BCG followed by three doses of M72 (2–5 μg), (iii) BCG followed by three doses of ABDsM72 (2–5 μg), or (iv) PBS as a control. Six weeks after the final immunization, the animals were aerosol-challenged with 2000 CFUs of *M. bovis* NCTC10772 using a GlasCol chamber (Glas-Col LLC, Terre Haute, IN, USA). We selected this vaccination protocol based on its demonstrated efficacy in our prior studies [18,20]. One month post-challenge, the mice were euthanized, and the lungs and spleens were harvested for bacterial quantification and other studies (see below).

Six weeks after the final immunization, mice were aerosol-challenged with 2000 CFUs of *M. bovis* NCTC10772 using a GlasCol chamber (Glas-Col LLC, Terre Haute, IN, USA). Four weeks post-challenge, the bacterial loads in lungs and spleens were quantified. Briefly, the tissues were homogenized (OMNI Polytron homogenizer, Kennesaw, GA, USA), and serial dilutions of the homogenates were plated on 7H10 agar supplemented with ADP, 10 μg/mL amphotericin B, and 125 μg/mL ampicillin to assess the number of CFUs after 5 weeks of incubation at 37 °C.

All animals used in this study were female to avoid sex-specific effects.

All mouse experiments were conducted in compliance with the regulations of the Institutional Animal Care and Use Committee (CICUAE) of INTA (approval code: 34/2023; approval date: 21 December 2023).

### 2.5. Lung Tissue Processing and Flow Cytometry Analysis

The inferior lobe of the right lung obtained from vaccinated/unvaccinated and *M. bovis*-challenged mice was dissected into 1 mm^3^ pieces using sterile surgical scissors. The tissue fragments underwent enzymatic digestion with 1 mg/mL collagenase D (Roche Diagnostics, Basel, Switzerland) in a 37 °C water bath for 30 min with periodic agitation. The digested tissue was mechanically dissociated through a 70 μm nylon mesh (BD Falcon, NY, USA) to obtain a single-cell suspension. Erythrocyte lysis was performed using ACK buffer (0.15 M NH_4_Cl, 10 mM KHCO_3_, 0.1 mM Na_2_EDTA) for 2 min at room temperature, followed by PBS washes.

The cells were resuspended in complete RPMI-1640 medium (supplemented with 20% heat-inactivated FBS and 1× antibiotic–antimycotic) and plated at a density of 2 × 10^6^ cells/well in 24-well plates. The cells were stimulated for 24 h with 20 μg/mL *M. bovis* Purified Protein Derivative from *M. bovis* (PPDb) in RPMI with 10% FBS and then stained with the following antibody panel: anti-CD4-PE (clone GK1.5), anti-KLRG1-FITC (clone 2F1), anti-CXCR3-PerCP/Cy5.5 (clone CXCR3-173), anti-PD-1-PerCP/Cy5.5 (clone 29F.1A12), and anti-CCR7-AF647 (clone 4B12) (all from BioLegend, San Diego, CA, USA).

Flow cytometry acquisition was performed using a BD FACScalibur system (Becton Dickinson, Franklin Lakes, NJ, USA), with the data analysis conducted using CellQuest Pro 3.3 software. The gating strategy followed established protocols [18] (Appendix A).

### 2.6. Histopathology

Tissue samples from the right lung apex, spleen, and liver were fixed in 10% neutral-buffered formalin and processed using conventional histological techniques. The specimens underwent sequential dehydration in graded ethanol solutions (70–100%), clearing in xylene, and embedding in paraffin at 56 °C. Sections that were 4–5 μm thick were prepared using a rotary microtome (Thermo Scientific HM 325) and stained with haematoxylin–eosin to observe the general tissue morphology and Ziehl–Neelsen stain for acid-fast bacilli detection. For the microscopic evaluation of the lung, liver, and spleen tissues, we assessed histopathological lesions based on the criteria established by Aguilar León et al. (2009) [21]. Macrophage aggregations (composed of macrophages and epithelioid cells) were quantified for each tissue section, while the following features were scored as present or absent: caseous necrosis within macrophage aggregations, calcium deposits (mineralization), Langhans giant cells, bronchus-associated lymphoid tissue (BALT), lymphocytic infiltrates, vascular congestion, and haemorrhage. The liver specimens were also evaluated for necrotic foci. All findings were systematically recorded.

### 2.7. Statistical Analysis

Quantitative data, including percentages of cell populations from flow cytometry, macrophage aggregation counts from histopathological examinations, and CFU measurements, were analysed using GraphPad Prism software (version 5.0, GraphPad Software, San Diego, CA, USA). Prior to analysis, data normality was assessed using the Shapiro–Wilk test, and outliers were detected via Grubbs’ test. Depending on the results, either ANOVA (with Bonferroni post hoc tests) or Kruskal–Wallis (followed by Dunn’s post hoc tests) was performed. A *p*-value < 0.05 was considered significant.

## 3. Results

### 3.1. Expression and Purification of M72 and ABDsM72

M72 is a chimeric protein comprising the C-terminal domain of PepA, the PPE18 protein, and the N-terminal domain of PepA [22]. A chimeric gene encoding this construct was generated as described in Section 2 (Figure 1C). Additionally, we designed a truncated variant, ABDsM72, in which the N-terminal domain of PPEP18 was replaced by a bovine albumin-binding domain (ABD: LAEAKVLANRELDKYGVSDFYKRLINKAKTVEGVEALKLHILAALP) (Figure 1C). Both recombinant proteins included an N-terminal hexahistidine (His_6_) tag for affinity purification. The full-length M72 antigen genetically fused to ABD was excluded from the assay due to insufficient expression levels.

The SDS-PAGE analysis (Figure 1A,B) confirmed the successful expression and purification of both constructs, with observed molecular weights of ~70 kDa for M72 and ~50 kDa for ABDsM72, which are consistent with their predicted sizes.

### 3.2. Evaluation of Protein Vaccines Combined with BCG for Control of M. bovis Organ Colonization and Histopathology Outcomes

All the vaccinated and challenged groups exhibited significantly reduced lung bacterial burdens (lower CFU counts) compared to the PBS controls (*p* < 0.001), with no significant differences among the vaccinated groups (Figure 2). However, only the BCG-M72 vaccination strategy significantly reduced splenic bacterial loads compared to the unvaccinated controls (*p* < 0.05) (Figure 2C).

Histopathological examination of mouse tissues revealed no significant differences between experimental groups in terms of either macrophage aggregation or the presence of lesions (Appendix A). One possible explanation for the lack of histopathology differences between the vaccinated and unvaccinated mice is that the analysis was performed during the early stage of lesion formation, a phase in which correlations with mycobacterial loads in organs are not typically observed. In fact, our previous studies found no correlation between the number of macrophage aggregations and CFU counts in mouse lungs [23].

### 3.3. Analysis of Lung Immune Responses Following BCG Prime-Protein Boost Vaccination and M. bovis Challenge

Given the established association between CXCR3+ lung-resident CD4+ T cells and protective immunity against mycobacteria [24,25,26], we focused on this cell population. Flow cytometry revealed that BCG-ABDsM72-vaccinated mice exhibited a significant increase in PPDb-responsive CD4+CXCR3+ T cells compared to all the other groups (Figure 3A).

KLRG1, a marker of terminal differentiation, is present in lung vasculature-associated cells but absent from parenchymal T cells [27]. We observed a significantly lower proportion of CD4+KLRG1+ T cells in BCG/M72-vaccinated mice compared to the unvaccinated controls (Figure 3B). Notably, the analysis of CD4+ T cells that were CXCR3+ and KLRG1− (indicative of parenchymal localization) revealed that the BCG-ABDsM72 group had the highest proportion of these cells among all the groups (Figure 3C). These findings suggest that BCG-ABDsM72 vaccination promotes the accumulation of lung-homing T cell subsets that are linked to enhanced anti-mycobacterial immunity, despite equivalent *M. bovis* CFU counts across all the vaccinated groups.

PD-1, another marker associated with TB protection, did not show significant differences in expression between the groups. Similarly, no differences were observed in the percentages of CCR7+ CD4+ T cells (Appendix A), a marker of central memory T cells [28].

We observed a negative correlation between the percentage of CD4+KLRG1−CXCR3+ T cells and lung CFU counts (Figure 4A). Conversely, CD4+KLRG1+ and CD4+PD1+ T cells exhibited a positive correlation with lung CFU counts (Figure 4B,C). The latter finding was unexpected given PD1’s established association with TB protection (Wykes and Lewin, 2018) [29].

Further analysis revealed no correlation between PD1 expression in lung parenchyma-resident (KLRG1−) central memory (CCR7+) CD4+ T cells and CFU counts, nor were there differences in the CD4+KLRG1−PD1+CCR7+ subset across the mouse groups (Appendix A). In addition, no significant changes in PD1+ lung-homing central memory T cells (KLRG1−CCR7+) were observed between the vaccinated and unvaccinated mice (Appendix A).

## 4. Discussion

Historically, the lack of commercial bTB vaccines has been attributed to their potential interference with standard diagnostic tests. However, recent advances in DIVA diagnostics—particularly those based on ESAT-6, CFP-10, and Rv3615c—have addressed this challenge [30]. Importantly, the vaccine formulations tested in this study should be compatible with a DIVA diagnostic approach as they rely on the immunodominant antigens ESAT-6, CFP-10, and Rv3615c—proteins not expressed by BCG. This key feature makes the BCG/M72 and BCG/ABDsM72 vaccines promising candidates for field evaluation in cattle.

From an economic perspective, even a 50% reduction in bTB cases through BCG vaccination could yield up to 75% indirect protection, substantially reducing the financial burden on producers who are currently mandated to cull infected animals [31]. Using a BCG prime-boost vaccination strategy, like the one evaluated in this study, protective efficacy exceeding 50% in cattle exposed to bTB is expected.

Among the candidate vaccines, M72/AS01E—a subunit vaccine comprising a fusion protein of *M. tuberculosis* antigens (PepA and PPE18) and the AS01E adjuvant system—has demonstrated 49.7% efficacy in preventing active TB in latently infected adults [10]. Earlier formulations of M72 conferred protection in mice and guinea pigs, although boosting BCG with M72/AS02A did not reduce mouse lung bacterial loads compared to BCG alone [32]. Skeiky and collaborators found that M72 formulated in AS01B elicited a stronger CD8+ T cell response (CTL and IFN-γ) than M72 formulated with AS02 in immunized C57BL/6 mice [22], highlighting the critical role of adjuvant selection in vaccine efficacy.

In the present study, we conjugated recombinant antigens to chitosan/alginate NCs, a decision based on our prior findings demonstrating that chitosan NCs loaded with the fusion protein H65 enhanced protection against *M. bovis* challenge in mice, as indicated by a reduced bacillary load in lung tissue [18]. The biodistribution and lymphatic transport of nanoparticles are critically dependent on their physicochemical properties, particularly their size and surface charge. Prior studies have demonstrated distinct size-dependent trafficking patterns. For instance, large particles (500–2000 nm) remain primarily at the injection site and interact with dendritic cells, while smaller nanoparticles (20–200 nm) are efficiently drained to lymph nodes where they are taken up by resident dendritic cells and macrophages [17]. Surface charge similarly impacts lymphatic uptake. In this study, the NCs exhibited a negative surface charge, attributable to the outer alginate coating. This characteristic may enhance lymphatic drainage, as negatively charged nanoparticles have been demonstrated to accumulate more efficiently in lymph nodes than their positively charged counterparts [33]. Notably, this alginate/chitosan composite design has previously shown immunological advantages. For instance, alginate-coated chitosan nanoparticles loaded with hepatitis A vaccine antigens elicited stronger humoral and cellular immune responses than uncoated chitosan nanoparticles [34]. In this study, we observed that boosting BCG with either M72 or a truncated M72 variant fused to an ABD domain similarly failed to enhance lung protection against *M. bovis* challenge. Several factors may explain the lack of efficacy of these booster vaccines in controlling *M. bovis* replication in the lungs, including the potentially low adjuvant capacity of the delivery system, the antigen dose administered, or simply a ceiling effect imposed by prior BCG vaccination in the murine model. However, the M72 boost significantly reduced bacterial dissemination to the spleen, suggesting a role in controlling systemic spread. In contrast, boosting with ABDsM72 did not enhance the control of splenic mycobacterial loads. This discrepancy may be due to the absence of the carboxyl-terminal region in ABDsM72, which in M72 may contain critical antigenic peptides necessary for an effective immune response. Notably, mice receiving BCG/ABDsM72 exhibited an elevated frequency of CD4+KLRG1-CXCR3+ T cells in the lungs, a population previously correlated with protection. While this finding aligns with the hypothesis that ABD enhances antigen drainage and dendritic cell uptake, the absence of a corresponding reduction in lung CFUs underscores the need for further mechanistic studies. The modest differences in CD4+KLRG1-CXCR3+ percentages between groups (despite statistical significance), combined with substantial intra-group variability in CFU counts, might explain why, in the group that received ABDsM72, the enhanced protective T cell response did not translate to a greater CFU reduction.

An important limitation of this study is that BALB/c mice do not develop granulomatous lesions or recapitulate the key aspects of *M. bovis* infection, such as transmission dynamics and chronic/latent infections. Therefore, evaluation in a more biologically relevant model (e.g., C3HeB/FeJ mice) is critical for a comprehensive assessment of the efficacy of the tested vaccine strategies. In addition, assessing both the long-term protective capacity and the induction of multifunctional T cell responses following vaccination will be essential to fully determine their effectiveness of these vaccines against *M. bovis* infection. Also, given the reported sex-based differences in tuberculosis vaccine responses in murine models [35,36], evaluating these vaccine candidates in male subjects is also necessary to ensure a complete understanding of their protective potential.

Interestingly, similar to our prior work, we found a link between higher proportions of CD4+KLRG1-CXCR3+ T cells and lower lung bacterial loads. However, despite showing increased levels of these T cells, the BCG/ABDsM72 group had lung CFU counts comparable to other groups. This discrepancy may reflect the robust protection conferred by BCG alone in mice, which could mask incremental improvements from prime-boost strategies. Longer-term studies or alternative animal models (e.g., guinea pigs) with greater susceptibility to TB may help to better elucidate the potential benefits of these regimens.

Central memory T cells (TCMs) are phenotypically distinct from effector memory T cells (TEMs) due to their expression of CCR7, a chemokine receptor that facilitates migration to secondary lymphoid organs via interactions with CCL19 and CCL21 [37]. In this study, we detected no significant differences in the frequencies of CCR7+ T cells in the lungs across the experimental groups. In contrast, our earlier research demonstrated a higher proportion of CD4+CCR7+ T cells in the lungs of vaccinated mice compared to unvaccinated controls following *M. bovis* challenge one year post-vaccination. These collective findings suggest that the expansion of CCR7+ T cells may depend on the interval of vaccination and pathogen exposure.

The dual role of PD-1 in immune regulation has gained increasing attention, particularly in the context of oncologic therapies and infectious diseases such as TB. In TB patients, elevated PD-1 expression has been observed on various immune cells, including CD4+ T cells, NK cells, neutrophils, and monocytes, suggesting a role in immune modulation [38]. However, despite its therapeutic value in cancer, anti-PD-1 therapy may disrupt the homeostasis of *M. tuberculosis*-specific T cells. This disruption can lead to extracellular matrix degradation, enhanced recruitment of monocytes and neutrophils, and dysregulation of key cytokines such as TNF-α—conditions that can potentially favour *M. tuberculosis* growth [39].

Studies using PD-1-deficient mice and observations from cancer patients undergoing PD-1 blockade underscore the non-redundant role of PD-1 in TB control. Paradoxically, high PD-1 expression on circulating T cells is a well-recognized hallmark of active TB disease [40,41], implying a context-dependent function. Our current findings support this duality: we observed a positive correlation between PD-1+CD4+ T cells and higher bacterial loads in patients with active TB [38]. This positive association suggests that PD-1 upregulation may reflect a host response to active infection rather than a protective immune mechanism induced by vaccination.

In conclusion, while prime-boost strategies using BCG and M72-based vaccines did not surpass BCG alone in preventing lung colonization, they reduced mycobacterial dissemination and modulated local immune responses.

Future studies should explore mucosal delivery routes and optimize antigen–NC formulations to enhance the efficacy of the vaccine candidates beyond the current findings.

## Figures and Tables

**Figure 1 pathogens-14-00592-f001:**
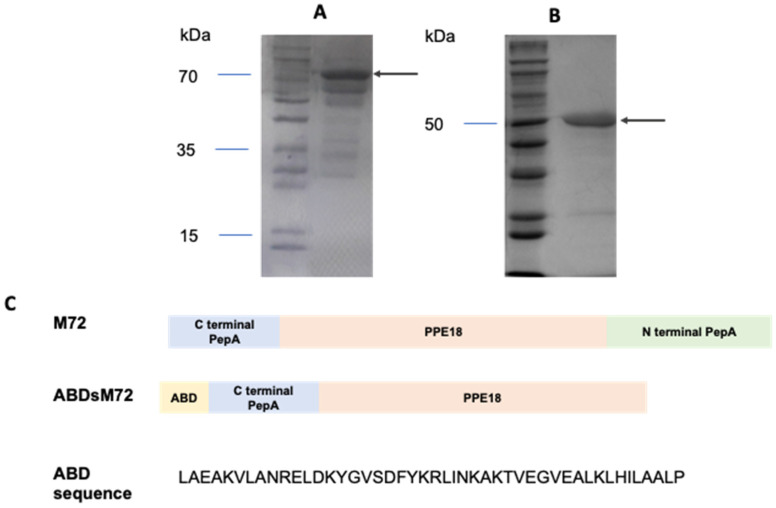
**Purification of recombinant M72 and ABDsM72.** Recombinant M72 (**A**) and ABDsM72 (**B**) were expressed in *E. coli* and purified using Ni-affinity chromatography. The purified proteins were resolved using a 12% SDS-PAGE gel and visualized by Coomassie blue staining. Arrows indicate the positions of the recombinant proteins. (**C**) Schematic representation of M72 and ABDsM72 and sequence of ABD.

**Figure 2 pathogens-14-00592-f002:**
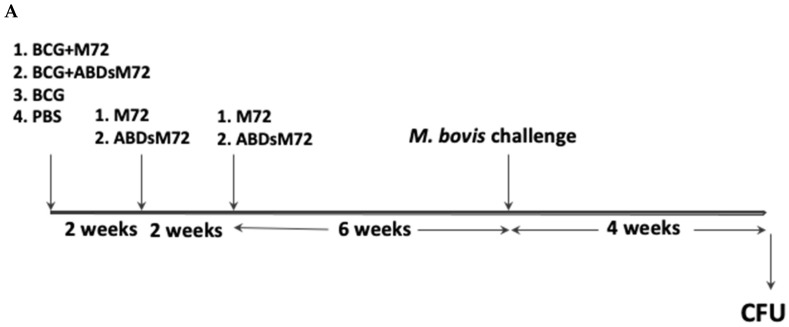
**Protection of vaccinated mice against *M. bovis* challenge.** (**A**) Schematic representation of the mouse vaccination trial for *M. bovis* infection. (**B**) Bacterial burden in lung tissue was quantified as colony-forming units (CFUs). Data are presented as mean ± standard error of the mean (SEM) and were analysed using one-way ANOVA followed by Bonferroni’s post hoc test for multiple pairwise comparisons (*n* = 7–8 animals/group) (*** *p* < 0.001). (**C**) Splenic bacterial loads were similarly determined and are shown as mean ± SEM. Statistical analysis was performed using the Shapiro–Wilk normality test and Kruskal–Wallis test followed by Dunn’s multiple comparison test (*n* = 7–8 animals/group) (* *p* < 0.05).

**Figure 3 pathogens-14-00592-f003:**
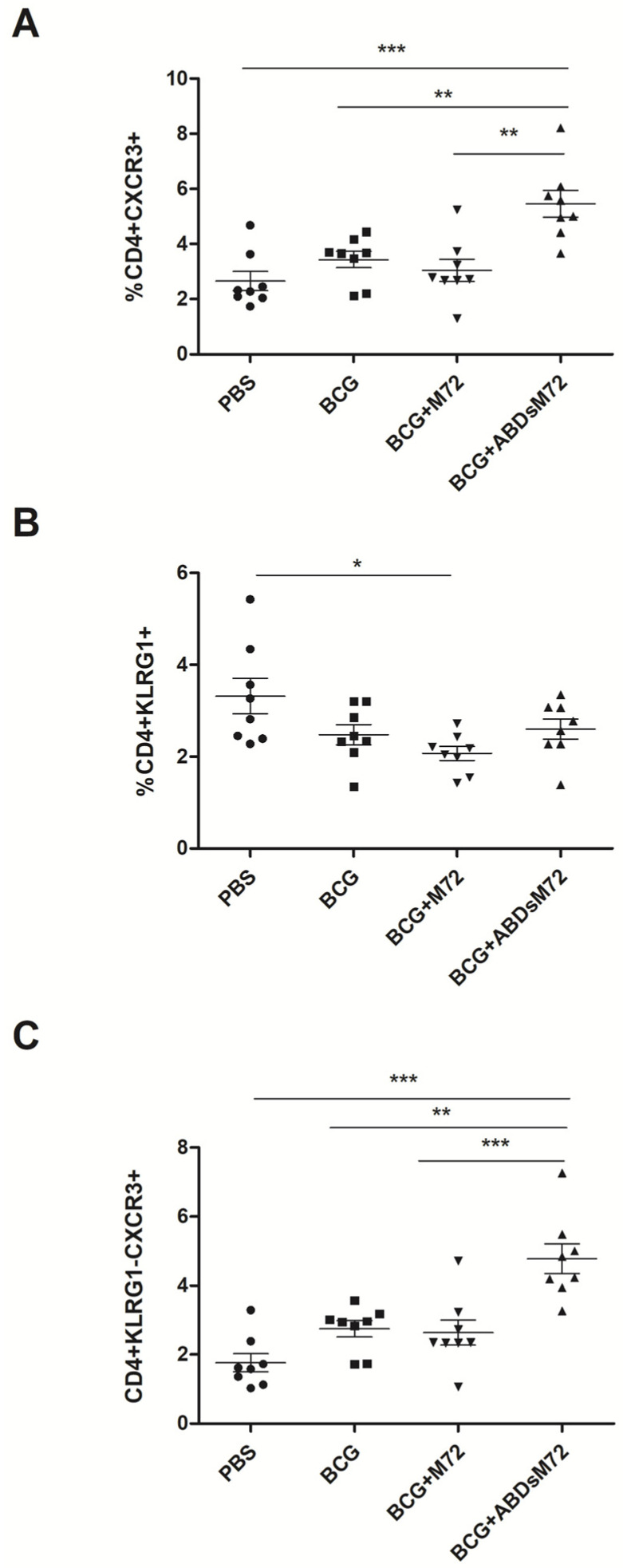
**Lung CD4+ T cell immunophenotyping following vaccination and *M. bovis* challenge.** Flow cytometric analysis of lymphocyte populations from lung homogenates showing percentages of (**A**) CXCR3+, (**B**) KLRG1+, and (**C**) CXCR3+KLRG1-subsets among CD4+ T cells. Bars represent mean values ± SEM for BCG-, M72-, and ABDsM72-vaccinated groups versus PBS controls (n = 8 animals/group). Significant differences were determined by one-way ANOVA with Bonferroni’s post hoc test (* *p* < 0.05; ** *p* < 0.01; *** *p* < 0.001). Complete gating strategy is presented in Appendix A.

**Figure 4 pathogens-14-00592-f004:**
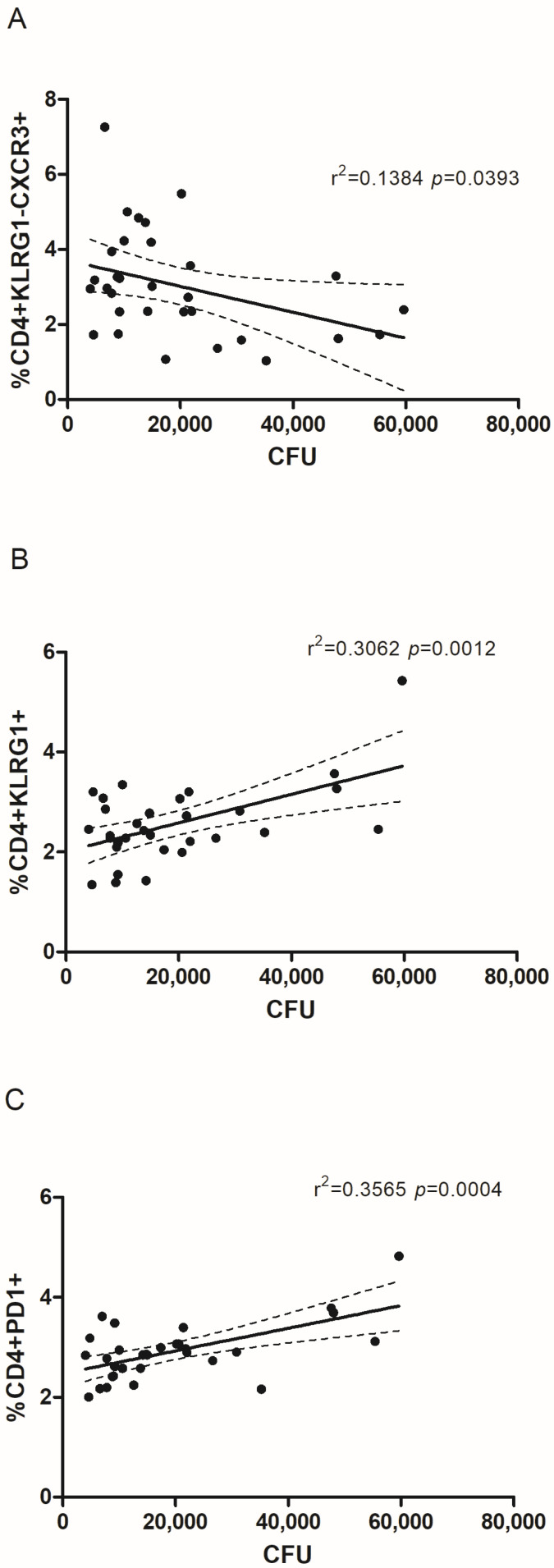
**Correlations between number of CFUs and CD4+ T cell populations.** A linear regression model was used to assess the relationship between lung CFU counts in vaccinated versus control animals and the proportions of (**A**) KLRG1-CXCR3+, (**B**) KLRG1+, and (**C**) PD1+ CD4+ T cell subsets. The regression line is represented by a solid line, with the dashed lines marking the 95% confidence intervals. The coefficient of determination (r^2^) and *p*-value from the regression analysis are provided.

## Data Availability

The original contributions presented in this study are included in the article/Appendix A. Further inquiries can be directed to the corresponding author.

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
