# Peer review of "M72 Fusion Proteins in Nanocapsules Enhance BCG Efficacy Against Bovine Tuberculosis in a Mouse Model"

_pathogens, 2025, doi:10.3390/pathogens14060592_

Round 1

Reviewer 1 Report

Comments and Suggestions for Authors

The authors evaluated a novel vaccination strategy combining BCG with either full-length M72 or a truncated M72 fused to a streptococcal albumin-binding domain (ABDsM72), both delivered via chitosan nanocapsules, in a murine M. bovis challenge model, and demonstrated its usefulnees protective effect.

I suggest to overcome in vivo model and demonstrate the same effect with an in vitro model and characterize the citokines profile.

Author Response

The authors evaluated a novel vaccination strategy combining BCG with either full-length M72 or a truncated M72 fused to a streptococcal albumin-binding domain (ABDsM72), both delivered via chitosan nanocapsules, in a murine M. bovis challenge model, and demonstrated its usefulnees protective effect.

I suggest to overcome in vivo model and demonstrate the same effect with an in vitro model and characterize the citokines profile.

Response: We appreciate the reviewer's valuable suggestion. However, there is currently no established in vitro model to assess vaccine efficacy against bovine tuberculosis.

Regarding cytokine profiling, we and others have observed that cytokine production does not serve as a correlate of protection. Instead, we assessed the proportion of CD4+ T cells expressing key markers in the lungs of challenged mice, as these T cell populations currently represent the best-established correlate of protection against tuberculosis (doi: 10.1016/j.vaccine.2024.04.055; doi: 10.4049/jimmunol.1400019; doi: 10.1128/IAI.00014-18; doi: 10.1038/mi.2016.70; doi: 10.1016/j.vetmic.2025.110371).

The manuscript has been thoroughly revised, and all modifications were highlighted.

Reviewer 2 Report

Comments and Suggestions for Authors

Comments for pathogens-3665602

Title: M72 fusion protein in chitosan nanocapsules enhances BCG efficacy against bovine tuberculosis in mouse model

General comments 

Abstract: This reports the use of chitosan nanocapsules to enhance the adjuvant effect of the M72 fusion protein, which has been shown to have a protective effect against M. tuberculosis when used in combination with the AS01E adjuvant in clinical trials, and evaluates it in an experimental infection system using BALB/C mice. This experiment appears to be worth attempting.

 A word might also be added about how chitosan microcapsules are said to enhance immune effects. 

Comments on individual parts

L27-28: It should be noted that the effect of inhibiting bacterial growth in the lungs was not significant compared to cases receiving BCG alone.

 I understand the importance of the spleen, but have any observations been made regarding the dynamics of immune cells and lesion formation in lymph nodes such as bronchial lymph nodes?

L28-30: Were these cells in the peripheral blood? What about the numbers of these cells in the spleen and lungs? Readers will be interested to know if this relates to differences in bacterial proliferation in different organs.

L39: Please make it a bit more specific and easy to understand.

L41: Wouldn't it be better to carry out regular testing? It would be inappropriate to use the onset of symptoms as the sole trigger for diagnostic testing. Are regular tests not carried out in the author's country?

L46: Would direct damages include the loss of value as beef cattle due to death or thinning of calves and adult cattle, and reduced milk yields of dairy cows? Furthermore, the assessor would like to know whether tuberculosis-infected cattle are allowed on farms in your country.

L57~L80: I think it is good that the overview of research on immune enhancement related to this study is presented.

L92~: The description is thorough and takes into consideration the reproducibility of the experiment.

L95: Is the VD3 you have here an activated form of VD? Please let me know since it is known that activated VD3 affects macrophage activation.

L124~: It is properly stated.

L160: Is it liquid nitrogen storage or freeze-dried bacteria?

L167-168: Is it necessary to write down the number of bacteria (antigen concentration) in the inoculated nanoparticles?

L176~177: It is better to mention the approval number given by CICUAE.

L178~: The methodology is well described.

L195~: The methodology is well described.

L205~: The methodology is well described.

L216~227: Since the description in 3.1 contains many items that should be described in the Materials and Methods, please avoid duplication and consider moving them to the Materials and Methods. All you need to write here is that the properties of the expressed recombinant protein matched the expected ones.

L236~242: Section 3.2 also contains many items that should be included in the Materials and Methods, so please avoid duplication and consider moving them to the Materials and Methods.

L247~249: Do not provide a discussion in the Results section, but rather in the Discussion section.

L291: Is it acceptable to call simple aggregation or local increase of macrophages "macrophage granuloma"? Please write the definition of granuloma used by the authors.

What do you mean by "the presence of lesions"? Are you referring to tuberculous nodules with surrounding connective tissue and lymphocytic infiltration?

There is a lot of duplication in what is written in the Materials and Methods section. Please keep what is written in the Materials and Methods section to a minimum and keep it concise so that the reader can understand it at a glance in the figures.

L268~272: Please include why you used that particular experimental method in the Materials and Methods section, rather than here. Please briefly describe only the results you obtained.

L280~283: Please explain this in your discussion, citing your results.

L294~299: This is not something you will discuss here, but rather in the discussion section, citing the results.

L303~305: This is not something we will discuss here, but rather in the discussion section, citing the results.

Discussion: Authors should not write a mini-review-like Discussion that describes what should be described in the introduction. In this section, authors should describe “the novel findings obtained from their experiments” in order of importance, with multiple references. This is a basic rule when writing a paper.

L314~323: What is written here should be written in the Introduction. If it is closely related to the significant results obtained, it is a good idea to cite and discuss the results.

L323-328: This is probably wishful thinking, but I wonder on what basis the authors' experimental results are based. Considering the differences in susceptibility and immune response of animals to tuberculosis infection, you would have to conduct a challenging experiment using calves before you could say anything like this.

L332~336: With these experimental results, can the authors expect to see results like those described in L326-336?

L337~342: It would be even better if the authors explained what they thought about this finding.

L355-359: First, state the results of the HO experiment which showed no significant differences, and then discuss the literature review and the author's interpretation of the data.

L368~370: Given the complexity of the immune system, it is understandable that working hypotheses do not always work as expected in experiments. I hope that further experiments will find an effective method.

For example, what would happen to the lesions and bacterial counts after vaccination if only BCG+ABDsM72 vaccination was administered without booster immunization with M72 or ABDsM72?

In addition, the M. bovis challenge was administered 6 weeks after the second booster vaccination, but how was the 6-week interval designed?

Also, have you conducted any tests in which the amount of BCG antigen in the first vaccination was changed?

Author Response

General comments

Abstract: This reports the use of chitosan nanocapsules to enhance the adjuvant effect of the M72 fusion protein, which has been shown to have a protective effect against M. tuberculosis when used in combination with the AS01E adjuvant in clinical trials, and evaluates it in an experimental infection system using BALB/C mice. This experiment appears to be worth attempting.

A word might also be added about how chitosan microcapsules are said to enhance immune effects.

R: The text was modified accordingly to reviewer suggestion: “Given the established efficacy of nanocapsule formulations as vaccine delivery systems, this study evaluates a novel immunization strategy combining BCG with either full-length M72 or a truncated M72 fused to a streptococcal albumin-binding domain (ABDsM72). Both antigens were encapsulated in chitosan/alginate nanocapsules and assessed in a murine M. bovis challenge model.”

Comments on individual parts

L27-28: It should be noted that the effect of inhibiting bacterial growth in the lungs was not significant compared to cases receiving BCG alone.

R: It was changed as following: “Priming with BCG followed by an M72 boost significantly improved splenic protection relative to BCG alone, but did not enhance pulmonary protection.”

 I understand the importance of the spleen, but have any observations been made regarding the dynamics of immune cells and lesion formation in lymph nodes such as bronchial lymph nodes?

R: lymph nodes were not analyzed in this study.

L28-30: Were these cells in the peripheral blood? What about the numbers of these cells in the spleen and lungs? Readers will be interested to know if this relates to differences in bacterial proliferation in different organs.

R: We analyzed only the lung immune response because the immune study was performed in the lungs of aerogenically challenged mice. We used the spleen only for CFU counts. We add “in the lungs of M. bovis-challenged mice” in the abstract to clarify this point.

L39: Please make it a bit more specific and easy to understand.

R: The text was modified as following: “In addition to cattle, bovine tuberculosis (bTB) infects other livestock species such as sheep, goats, and pigs, increasing its detrimental effects on animal health and agricultural productivity.”

L41: Wouldn't it be better to carry out regular testing? It would be inappropriate to use the onset of symptoms as the sole trigger for diagnostic testing. Are regular tests not carried out in the author's country?

R: Animals are tested once a year; therefore, the absence of early clinical signs makes detection between annual tests difficult. To avoid confusion, we removed the text from the discussion, as it is not relevant to this study

L46: Would direct damages include the loss of value as beef cattle due to death or thinning of calves and adult cattle, and reduced milk yields of dairy cows? Furthermore, the assessor would like to know whether tuberculosis-infected cattle are allowed on farms in your country.

R: In general, beef cattle do not die from tuberculosis, and losses due to carcass condemnation are not significant. The main loss comes from the early culling of animals, primarily in dairy herds. Milk production is also affected when cows have tuberculosis. In Argentina, animals that test positive to the official test of bTB are required to be sent to slaughter.

We include the following midfication: “(e.g. reduced milk productivity and early culling of animals in dairy herds).

We also include this text: “In Argentina, bTB is a notifiable disease, and animals that react positively to the official test are required to be slaughtered. (https://www.argentina.gob.ar/normativa/nacional/resoluci%C3%B3n-128-2012-195314).”

L57~L80: I think it is good that the overview of research on immune enhancement related to this study is presented.

L92~: The description is thorough and takes into consideration the reproducibility of the experiment.

L95: Is the VD3 you have here an activated form of VD? Please let me know since it is known that activated VD3 affects macrophage activation.

R; es, the VD3 was used in its activated form. We included the catalog number.

L124~: It is properly stated.

L160: Is it liquid nitrogen storage or freeze-dried bacteria?

R: The text now reads: “Fresh cultures of BCG and M. bovis NCTC10772 (the challenge strain) were processed as previously described [19] to prevent clumping prior to use. “

We found that is not adequate to infect mice with mycobacteria that were storage frozen because most of the bacteria are not viable.

L167-168: Is it necessary to write down the number of bacteria (antigen concentration) in the inoculated nanoparticles?

R: The antigen concentrations in the NCs were included (“Successful antigen loading into the NCs was confirmed by SDS-PAGE analysis (Fig. S1). The amounts of the recombinant proteins per microgram of NCs were 0.0018-0.0011mg and 0.0044-0.0026 mg for M72 and ABDsM72, respectively.”)

L176~177: It is better to mention the approval number given by CICUAE.

R: It was included as following: “(Approval Code: 34/2023, Approval Date: 21/12/2023).”

L178~: The methodology is well described.

L195~: The methodology is well described.

L205~: The methodology is well described.

L216~227: Since the description in 3.1 contains many items that should be described in the Materials and Methods, please avoid duplication and consider moving them to the Materials and Methods. All you need to write here is that the properties of the expressed recombinant protein matched the expected ones.

R: We eliminated the redundant information (“Protein expression was carried out in E. coli BL21(DE3) cells, followed by affinity purification using immobilized metal ion chromatography (IMAC) with Ni-NTA resin.”).

L236~242: Section 3.2 also contains many items that should be included in the Materials and Methods, so please avoid duplication and consider moving them to the Materials and Methods.

R: The following text was moved to M&M: “The experimental groups were: (i) BCG alone (1x10e5 CFU), (ii) BCG followed by three doses of M72 (2-5 mg), (iii) BCG followed by three doses of ABDsM72 (2-5 mg), or (iv) PBS as a control.”

L247~249: Do not provide a discussion in the Results section, but rather in the Discussion section.

R: The following text was eliminated. “These findings suggest that while all vaccination approaches effectively controlled pulmonary infection, the BCG-M72 combination offered superior protection against systemic dissemination, as evidenced by reduced splenic colonization.”

L291: Is it acceptable to call simple aggregation or local increase of macrophages "macrophage granuloma"? Please write the definition of granuloma used by the authors.

R: We have changed “macrophage granuloma” by “macrophage aggregation” to avoid confusions.

What do you mean by "the presence of lesions"? Are you referring to tuberculous nodules with surrounding connective tissue and lymphocytic infiltration?

R: we added the following text to describe the features of the lesions evaluated: “For microscopic evaluation of lung, liver, and spleen tissues, we assessed histopathological lesions based on the criteria established by Aguilar León et al. (2009) “.

There is a lot of duplication in what is written in the Materials and Methods section. Please keep what is written in the Materials and Methods section to a minimum and keep it concise so that the reader can understand it at a glance in the figures.

R: We eliminated the following text from figure 2: “(A) Four groups of eight mice each were included in the study. Three groups were immunized subcutaneously with BCG, the recombinant fusion protein M72, or the albumin-binding domain-fused sM72 (ABDsM72). A fourth group received phosphate-buffered saline (PBS) and served as the negative control. Thirty days after the final immunization, all animals were challenged via the aerosol route with 2,000 CFU of virulent M. bovis.”

L268~272: Please include why you used that particular experimental method in the Materials and Methods section, rather than here. Please briefly describe only the results you obtained.

R: We eliminated the following text “We analysed lung-localized immune responses upon in vitro stimulation with Protein Purified Derivative from M. bovis (PPDb) in vaccinated and unvaccinated mice following aerosol challenge with virulent M. bovis.” from the result section and included the following in M&M: “The inferior lobe of the right lung obtained from vaccinated/unvaccinated and M. bovis challenged mice…”

L280~283: Please explain this in your discussion, citing your results.

R: This text was eliminated to avoid redundancy.

L294~299: This is not something you will discuss here, but rather in the discussion section, citing the results.

R: The text was modified accordingly to reviewer´s suggestion.

L303~305: This is not something we will discuss here, but rather in the discussion section, citing the results.

R: The text was modified accordingly to reviewer´s suggestion.

Discussion: Authors should not write a mini-review-like Discussion that describes what should be described in the introduction. In this section, authors should describe “the novel findings obtained from their experiments” in order of importance, with multiple references. This is a basic rule when writing a paper.

L314~323: What is written here should be written in the Introduction. If it is closely related to the significant results obtained, it is a good idea to cite and discuss the results.

R: We eliminated the paragraph from the discussion.

L323-328: This is probably wishful thinking, but I wonder on what basis the authors' experimental results are based. Considering the differences in susceptibility and immune response of animals to tuberculosis infection, you would have to conduct a challenging experiment using calves before you could say anything like this.

R: We acknowledge the reviewer’s concern that the absence of cattle assays limits our ability to conclusively demonstrate compatibility with a DIVA  strategy. However, our vaccine candidate shares a key feature with BCG—namely, the absence of the DIVA target antigen. Since BCG has been empirically shown in cattle trials to be compatible with DIVA testing, we anticipate a similar outcome for our construct. In response to this comment, we have revised the phrasing from "is compatible" to "should be compatible" to reflect this inference. We have retained this discussion as requested by Reviewer 3, but with the added clarification to ensure scientific caution.

L332~336: With these experimental results, can the authors expect to see results like those described in L326-336?

R: This text was included in the discussion: “Using a BCG prime-boost vaccination strategy, as evaluated in this study, protective efficacy exceeding 50% in cattle exposed to bTB is expected.”

L337~342: It would be even better if the authors explained what they thought about this finding.

R: We included the following paragraph: “The modest differences in CD4+KLRG1-CXCR3+ percentages between groups (despite statistical significance), combined with substantial intra-group variability in CFU counts, might explain why in the group that received ABDsM72, the enhanced protective T cell response didn't translate to greater CFU reduction.”

L355-359: First, state the results of the HO experiment which showed no significant differences, and then discuss the literature review and the author's interpretation of the data.

R: We apologize to this reviewer for not understanding the criticism and what we should modify according to his/her suggestion.

L368~370: Given the complexity of the immune system, it is understandable that working hypotheses do not always work as expected in experiments. I hope that further experiments will find an effective method.

For example, what would happen to the lesions and bacterial counts after vaccination if only BCG+ABDsM72 vaccination was administered without booster immunization with M72 or ABDsM72?

In addition, the M. bovis challenge was administered 6 weeks after the second booster vaccination, but how was the 6-week interval designed?

Also, have you conducted any tests in which the amount of BCG antigen in the first vaccination was changed?

R: We agree that changing the vaccination and challenge schedules could yield different results. However, our approach was based on a previously published vaccination protocol that demonstrated effectiveness. This clarification was added to the Results section. Furthermore, we included in the Discussion the limitations of this study and the additional parameters that would need to be evaluated in future experiments:

 “An important limitation of this study is that BALB/c mice do not develop granulomatous lesions or recapitulate key aspects of M. bovis infection, such as transmission dynamics and chronic/latent infection. Therefore, evaluation in a more biologically relevant model (e.g., C3HeB/FeJ mice) is critical for a comprehensive assessment of the efficacy of the tested vaccine strategies. In addition, assessing both the long-term protective capacity and the induction of multifunctional T cell responses following vaccination will be essential to fully determine their effectiveness of these vaccines against M. bovis infection. Also, given the reported sex-based differences in tuberculosis vaccine responses in murine models [36,37], evaluating these vaccine candidates in male subjects is also necessary to ensure a complete understanding of their protective potential.”

Reviewer 3 Report

Comments and Suggestions for Authors

The manuscript addresses an important gap in bovine tuberculosis (bTB) vaccine development, employs innovative delivery strategies, and provides valuable immunological insights. However, several aspects require clarification, methodological refinement, and deeper discussion to enhance scientific rigor and impact.

  1. The study is well-motivated, leveraging recent advances in subunit vaccine design, nanocarrier technology, and immunological correlates of protection. The focus on chitosan nanocapsules as delivery vehicles is justified by their biocompatibility, adjuvant properties, and ability to enhance both humoral and cellular responses. The fusion of M72 with an albumin-binding domain (ABD) to promote lymphatic targeting is innovative and supported by emerging literature on albumin-hitchhiking strategies.
  • While the rationale is compelling, the introduction could better articulate the specific gaps in bTB vaccine development that this study addresses. For instance, the challenges of DIVA (Differentiating Infected from Vaccinated Animals) compatibility and the limitations of murine models for bTB pathology are mentioned but not fully developed. The authors should contextualize their approach within the broader landscape of bTB vaccine candidates and clarify how their strategy overcomes or complements existing limitations.
  1. The methods for recombinant protein expression, nanocapsule preparation, and antigen loading are described in detail. The use of solvent displacement for nanocapsule synthesis and the subsequent loading of M72 or ABDsM72 is appropriate and follows established protocols. Dynamic light scattering (DLS) and scanning electron microscopy (SEM) are used to characterize particle size, zeta potential, and morphology, with supplementary data provided. The methodological rigor in nanocapsule characterization is commendable. The authors report hydrodynamic radius, zeta potential, and SEM images for both unloaded and antigen-loaded nanocapsules, ensuring reproducibility and transparency. The inclusion of cryoprotectants and freeze-drying steps is appropriate for vaccine stability.
  • The degree of antigen loading (i.e., encapsulation efficiency and antigen release kinetics) is not reported.
  • Quantitative data on how much antigen is associated with the nanocapsules, and how it is released over time, are critical for interpreting immunogenicity and efficacy. The authors should include or reference data on antigen loading efficiency and in vitro release profiles.
  1. The physicochemical properties of chitosan (molecular weight, degree of deacetylation) are specified, but the potential variability in chitosan source and batch-to-batch consistency should be discussed, as these factors can influence immunogenicity and reproducibility.
  • The rationale for the chosen nanocapsule size range (333–341 nm) should be elaborated. While particles in this size range are generally suitable for uptake by antigen-presenting cells, smaller particles (<200 nm) may have different biodistribution and lymphatic targeting properties. The authors should discuss how particle size may affect vaccine performance.
  1. The cloning, expression, and purification of M72 and ABDsM72 in  coliare described in detail, including the use of affinity chromatography and quality control by SDS PAGE. The fusion of ABD to M72 is justified by its potential to enhance lymphatic uptake. The use of standard molecular biology techniques and clear reporting of construct design, expression conditions, and purification steps enhances reproducibility. The inclusion of both full-length and ABD-fused M72 allows for comparative analysis of antigen delivery strategies.
  • The immunogenicity of the ABD domain itself should be discussed. While ABD fusion is intended to enhance delivery, it may also introduce new epitopes or alter antigen processing. The authors should clarify whether ABD alone was tested for immunogenicity or included as a control.
  • Endotoxin removal and quantification are not mentioned. Recombinant proteins expressed in  colican be contaminated with endotoxins, which may confound immunological readouts. The authors should report endotoxin levels and describe any removal procedures.
  1. The use of female BALB/c mice, subcutaneous immunization with BCG, and subsequent boosting with M72 or ABDsM72 nanovaccines is appropriate for a preclinical evaluation. The challenge with virulent  bovisand assessment of bacterial loads in lungs and spleens are standard endpoints. The experimental design includes appropriate control groups (PBS, BCG alone), sufficient group sizes (n=8), and well-defined immunization and challenge schedules. The use of both pulmonary and systemic bacterial burden as outcomes provides a comprehensive assessment of vaccine efficacy.
  • The choice of BALB/c mice, while common, has inherent limitations for modeling human or bovine TB pathology. Mice do not develop the full spectrum of granulomatous lesions seen in natural hosts, nor do they recapitulate transmission dynamics or chronic/latent infection. The authors should acknowledge these limitations and discuss their implications for translational relevance.
  • The route of immunization (subcutaneous) and challenge (aerosol) are appropriate, but the rationale for booster timing (2 and 4 weeks post-primary vaccination) should be justified based on immunological kinetics or prior studies.
  • The duration of follow-up post-challenge (4 weeks) may be insufficient to capture long-term protection or memory responses. The authors should discuss whether longer-term studies are planned or warranted.
  1. Flow cytometry is used to characterize lung-resident CD4+ T cell subsets, focusing on markers associated with protective immunity (CXCR3, KLRG1, PD-1, CCR7). Histopathology assesses granuloma formation and tissue lesions. The immunophenotyping strategy is robust, employing established markers and appropriate gating controls (FMO). The focus on lung-resident T cell subsets is justified by their association with protection in TB models. The use of both functional (CFU reduction) and mechanistic (cellular correlates) endpoints strengthens the study.
  • The selection of markers (CXCR3, KLRG1, PD-1, CCR7) is appropriate, but the lack of functional assays (e.g., cytokine production, proliferation, cytotoxicity) limits the mechanistic interpretation. The authors should consider including or discussing functional readouts to complement phenotypic data.
  • The statistical analysis is generally sound, but the use of multiple comparison corrections and reporting of exact p-values should be clarified. The authors should specify whether data met assumptions for parametric tests and provide effect sizes where possible.
  • The histopathological scoring system is not fully described. Quantitative criteria for granuloma assessment, lesion grading, and observer blinding should be detailed to ensure objectivity and reproducibility.
  1. The authors report that both unloaded and antigen-loaded chitosan/alginate nanocapsules exhibit a rounded morphology with a dense oily core, as confirmed by SEM12. DLS measurements indicate hydrodynamic radii of 333–341 nm and zeta potentials ranging from -25 to -38 mV, consistent with stable colloidal suspensions. The detailed characterization of nanocapsule morphology and surface charge supports the integrity and stability of the vaccine formulations. The use of supplementary figures and tables enhances data transparency.
  • The absence of antigen loading efficiency and release kinetics data, as noted above, is a significant gap. These parameters are essential for correlating nanocapsule properties with immunogenicity and efficacy.
  1. SDS-PAGE analysis confirms the expected molecular weights of M72 (~70 kDa) and ABDsM72 (~40 kDa), with successful purification from  colilysates. The use of His-tag affinity purification is standard and effective. The clear demonstration of protein expression and purity supports the validity of subsequent immunization experiments.
  • The potential for residual host cell proteins or contaminants should be addressed, particularly regarding immunogenicity and safety.
  1. All vaccinated groups (BCG, BCG+M72, BCG+ABDsM72) exhibit significantly reduced lung CFU counts compared to PBS controls, with no significant differences among vaccinated groups. However, only the BCG+M72 group shows a significant reduction in splenic CFUs, suggesting enhanced control of systemic dissemination. The demonstration of significant protection against pulmonary infection by all vaccine regimens is robust and consistent with prior studies. The additional reduction in splenic dissemination by the M72 boost is a notable finding.
  • The lack of enhanced lung protection by booster regimens compared to BCG alone is a key limitation, consistent with other studies in nonhuman primates and mice. The authors should discuss possible reasons, such as the ceiling effect of BCG in murine models, suboptimal antigen dosing, or insufficient adjuvant potency.
  • The absence of long-term follow-up limits conclusions about durability of protection. The authors should consider or discuss extended challenge studies.
  • The use of only female mice may limit generalizability, as sex differences in TB immunity are documented. The authors should acknowledge this limitation.
  1. Flow cytometry reveals that BCG+ABDsM72 vaccination significantly increases the frequency of lung CD4+CXCR3+KLRG1- T cells, a subset associated with protective immunity2. PD-1 expression and CCR7+ central memory T cells do not differ significantly among groups. Correlation analysis shows a negative association between lung CFUs and CD4+CXCR3+KLRG1- T cells, but a positive association with KLRG1+ and PD-1+ subsets. The identification of lung-homing, parenchyma-resident CD4+ T cell subsets as correlates of protection is a valuable mechanistic insight. The use of regression analysis to link immune phenotypes with bacterial burden strengthens the conclusions.
  • The lack of functional assays (e.g., cytokine production, recall responses) limits the mechanistic depth. The authors should consider including ELISPOT, intracellular cytokine staining, or proliferation assays to assess T cell functionality.
  • The unexpected positive correlation between PD-1+ T cells and bacterial burden warrants deeper discussion. The dual role of PD-1 in TB protective versus exhaustion should be explored in light of recent literature.
  • The absence of significant changes in CCR7+ central memory T cells should be discussed in the context of vaccine-induced memory and tissue residency.
  1. No significant differences are observed among groups in granuloma formation, macrophage infiltration, or lesion presence. This finding is consistent with the limitations of murine models for recapitulating human or bovine TB pathology. The inclusion of histopathological analysis provides a complementary endpoint to bacterial burden and immune phenotyping.
  • The lack of detailed quantitative scoring and representative images limits interpretability. The authors should provide more granular data and discuss the limitations of murine pathology for modeling bTB.
  1. The study demonstrates that chitosan nanocapsule-formulated M72 and ABDsM72 antigens, when used to boost BCG vaccination, do not enhance pulmonary protection against  bovisin mice but do reduce systemic dissemination (splenic CFUs) and modulate lung-resident T cell populations. The increase in CD4+CXCR3+KLRG1- T cells in the BCG+ABDsM72 group supports the potential of albumin-hitchhiking strategies for improving lymphatic targeting and immune priming. The authors provide a balanced discussion, acknowledging the limitations of their findings and situating them within the broader context of TB and bTB vaccine development. The recognition that BCG’s robust protection in mice may mask incremental improvements from booster regimens is insightful.
  • The discussion could be strengthened by a more critical appraisal of the translational relevance of murine findings. The limitations of mouse models for bTB, including differences in granuloma structure, chronicity, and immune responses, should be explicitly addressed.
  • The potential implications for DIVA compatibility and field deployment in cattle are mentioned but not fully developed. The authors should elaborate on how their vaccine strategy could be integrated with current diagnostic and control programs.
  • The mechanistic basis for the observed reduction in splenic dissemination by M72 boosting, but not ABDsM72, should be explored. Possible explanations include differences in antigen processing, trafficking, or immune activation.
  • The lack of significant differences in histopathology and memory T cell responses should be discussed in the context of vaccine-induced immunity and protection.
  1. The findings are consistent with prior studies showing that BCG prime–subunit boost regimens (including M72/AS01E) often fail to enhance pulmonary protection in animal models, despite inducing robust immune responses. The partial efficacy of M72/AS01E in human trials (approximately 50% protection) underscores the challenge of translating preclinical findings to the clinic. The authors appropriately reference recent trials and reviews, situating their work within the evolving landscape of TB vaccine research.
  • A more detailed comparison with other adjuvant and delivery strategies (e.g., viral vectors, alternative nanocarriers) would enhance the discussion. The potential advantages and disadvantages of chitosan nanocapsules relative to other platforms should be articulated.
  1. The authors suggest that longer-term studies, alternative animal models (e.g., guinea pigs, cattle), and mechanistic investigations are warranted. These recommendations are appropriate and align with the study’s limitations.
  • The authors should outline specific experimental plans for addressing key gaps, such as testing in natural hosts, evaluating mucosal delivery routes, or optimizing antigen/adjuvant combinations.
  • The potential for chitosan nanocapsules to enhance mucosal immunity, given their mucoadhesive properties, should be discussed in greater detail.
  1. The study employs appropriate controls, randomization, and blinding where applicable. The statistical analyses are generally sound, though additional details on data distribution, effect sizes, and corrections for multiple comparisons are needed. The use of multiple complementary endpoints (CFU counts, flow cytometry, histopathology) enhances internal validity. The consistency of findings with prior studies supports the reliability of the results.
  • Potential confounders, such as endotoxin contamination, batch variability in nanocapsule preparation, and sex-specific effects, are not fully addressed. The authors should report measures taken to control for these variables.
  1. The generalizability of findings to natural hosts (cattle), other vaccine platforms, and field conditions is limited by the use of a murine model and controlled laboratory settings. The authors appropriately acknowledge these limitations.
  • The authors should discuss the steps needed to translate their findings to cattle or other relevant species, including dose scaling, safety assessment, and regulatory considerations.
  • The potential for chitosan-based vaccines to interfere with or complement existing diagnostic tests (e.g., DIVA assays) should be elaborated.

Revision and comments:

To improve the manuscript’s clarity, rigor, and impact, the following specific revisions are requested:

  1. Include quantitative data on antigen encapsulation efficiency and in vitro release kinetics from chitosan nanocapsules. Discuss how these parameters may influence immunogenicity and efficacy.
  2. Incorporate functional assays (e.g., cytokine production by ELISPOT or intracellular staining, proliferation, cytotoxicity) to complement phenotypic T cell analysis. If not feasible, discuss this limitation explicitly.
  3. Provide detailed quantitative scoring criteria for granuloma and lesion assessment, representative histological images, and information on observer blinding.
  4. Discuss the implications of the short follow-up period and consider extending the duration of post-challenge observation in future studies.
  5. Report the age of mice at immunization and challenge. Discuss the potential impact of using only female mice and consider including both sexes in future experiments.
  6. Describe measures taken to ensure consistency in chitosan and antigen preparations. Report endotoxin levels in recombinant protein batches and steps taken for removal.
  7. Elaborate on how the vaccine candidates could be integrated with current or future DIVA diagnostic assays. Consider experimental evaluation if feasible.
  8. Discuss the steps required to translate findings to cattle or other relevant species, including dose scaling, safety, and regulatory considerations.
  9. Provide a more detailed comparison of chitosan nanocapsules with other vaccine delivery platforms (e.g., viral vectors, liposomes, other polymers), highlighting advantages and potential challenges.
  10. Given the mucoadhesive properties of chitosan, discuss the potential for mucosal (e.g., intranasal) delivery routes and their relevance for bTB control.
  11. Ensure that all figures and tables are clearly labeled, with appropriate statistical annotations and sample sizes indicated.
  12. Expand the discussion of how the findings align with or diverge from recent studies on BCG prime–boost regimens, M72/AS01E efficacy, and nanocarrier-based vaccines.

In summary, the manuscript presents a well-conceived and methodologically sound investigation of chitosan nanocapsule-formulated M72 and ABDsM72 antigens as boosters to BCG vaccination in a murine model of M. bovis infection. The study’s strengths include its innovative delivery platform, comprehensive immunological analysis, and relevance to bTB control. However, several limitations, particularly regarding antigen loading, functional immunity, model system constraints, and translational relevance, should be addressed to strengthen the scientific rigor and impact of the work.

Comments on the Quality of English Language

The quality of English language in the manuscript is generally high. The text is clear, precise, and professionally written, with appropriate use of scientific terminology and a logical structure throughout. Sentences are well-constructed, and the narrative flows smoothly between sections. The abstract, introduction, methods, results, and discussion are all articulated in a manner consistent with international standards for scientific writing.

Minor issues has been observed such as occasional long or complex sentences could be further simplified for clarity, especially for readers less familiar with the subject matter.

Next, Some typographical inconsistencies (e.g., spacing around units or symbols) are present but do not impede understanding.

Last, a few minor formatting issues (such as line breaks or section transitions) appear to result from document conversion rather than language errors.

Overall, the manuscript does not contain significant grammatical or stylistic errors, and the English language is of publication quality. Only minor editorial polishing may be needed prior to final acceptance.

Author Response

The manuscript addresses an important gap in bovine tuberculosis (bTB) vaccine development, employs innovative delivery strategies, and provides valuable immunological insights. However, several aspects require clarification, methodological refinement, and deeper discussion to enhance scientific rigor and impact.

We sincerely appreciate the reviewer’s thoughtful and constructive feedback on our manuscript. We have carefully addressed all of their concerns, and we believe the revisions have significantly strengthened the paper. Below, we provide a point-by-point response to each comment.

  1. The study is well-motivated, leveraging recent advances in subunit vaccine design, nanocarrier technology, and immunological correlates of protection. The focus on chitosan nanocapsules as delivery vehicles is justified by their biocompatibility, adjuvant properties, and ability to enhance both humoral and cellular responses. The fusion of M72 with an albumin-binding domain (ABD) to promote lymphatic targeting is innovative and supported by emerging literature on albumin-hitchhiking strategies.
  • While the rationale is compelling, the introduction could better articulate the specific gaps in bTB vaccine development that this study addresses. For instance, the challenges of DIVA (Differentiating Infected from Vaccinated Animals) compatibility and the limitations of murine models for bTB pathology are mentioned but not fully developed. The authors should contextualize their approach within the broader landscape of bTB vaccine candidates and clarify how their strategy overcomes or complements existing limitations. R: To address this concern we included the following sentences in the introduction: “In this regard, the antigens ESAT-6, CFP-10, and Rv3615c—used as a cocktail or as a fused protein—have demonstrated diagnostic performance comparable to PPDb in intradermal tuberculin tests [9]. They offer similar sensitivity while improving specificity by avoiding cross-reactivity with non-tuberculous mycobacteria or BCG vaccination, as these antigens are absent from BCG strains [10].”

“Although mice do not fully replicate tuberculosis as it occurs in natural hosts, they remain the most widely used animal model for preclinical testing TB vaccines due to their manageable size, availability of well-characterized immune tools, and the diversity of genetically modified strains. ”

  1. The methods for recombinant protein expression, nanocapsule preparation, and antigen loading are described in detail. The use of solvent displacement for nanocapsule synthesis and the subsequent loading of M72 or ABDsM72 is appropriate and follows established protocols. Dynamic light scattering (DLS) and scanning electron microscopy (SEM) are used to characterize particle size, zeta potential, and morphology, with supplementary data provided. The methodological rigor in nanocapsule characterization is commendable. The authors report hydrodynamic radius, zeta potential, and SEM images for both unloaded and antigen-loaded nanocapsules, ensuring reproducibility and transparency. The inclusion of cryoprotectants and freeze-drying steps is appropriate for vaccine stability.
  • The degree of antigen loading (i.e., encapsulation efficiency and antigen release kinetics) is not reported. 

R: Successful antigen loading into the nanocapsules was confirmed by SDS-PAGE analysis. We included a new supplementary figure.

  • Quantitative data on how much antigen is associated with the nanocapsules, and how it is released over time, are critical for interpreting immunogenicity and efficacy. The authors should include or reference data on antigen loading efficiency and in vitro release profiles

R: We have now included quantitative data on protein loading efficiency in the nanocapsules. While we did not perform in vitro antigen release assays, as this study specifically focused on evaluating short-term protective efficacy, we acknowledge that characterization of antigen release kinetics would be essential for assessing long-term vaccine performance.

  1. The physicochemical properties of chitosan (molecular weight, degree of deacetylation) are specified, but the potential variability in chitosan source and batch-to-batch consistency should be discussed, as these factors can influence immunogenicity and reproducibility.
  • The rationale for the chosen nanocapsule size range (333–341 nm) should be elaborated. While particles in this size range are generally suitable for uptake by antigen-presenting cells, smaller particles (<200 nm) may have different biodistribution and lymphatic targeting properties. The authors should discuss how particle size may affect vaccine performance.

R: We have addressed this concern by including the following text in the discussion: “The biodistribution and lymphatic transport of nanoparticles are critically dependent on their physicochemical properties, particularly size and surface charge. Prior studies have demonstrated distinct size-dependent trafficking patterns. For instance, large particles (500–2000 nm) remain primarily at the injection site and interact with dendritic cells, while smaller nanoparticles (20–200 nm) are efficiently drained to lymph nodes, where they are taken up by resident dendritic cells and macrophages [18]. Surface charge similarly impacts lymphatic uptake. In this study, the nanocapsules exhibited a negative surface charge, attributable to the outer alginate coating. This characteristic may enhance lymphatic drainage, as negatively charged nanoparticles have demonstrated to accumulate more efficiently in lymph nodes than their positively charged counterparts [34]. Notably, this alginate/chitosan composite design has previously shown immunological advantages. For instance, alginate-coated chitosan nanoparticles loaded with hepatitis A vaccine antigens elicited stronger humoral and cellular immune responses than uncoated chitosan nanoparticles [35].”

  1. The cloning, expression, and purification of M72 and ABDsM72 in coliare described in detail, including the use of affinity chromatography and quality control by SDS PAGE. The fusion of ABD to M72 is justified by its potential to enhance lymphatic uptake. The use of standard molecular biology techniques and clear reporting of construct design, expression conditions, and purification steps enhances reproducibility. The inclusion of both full-length and ABD-fused M72 allows for comparative analysis of antigen delivery strategies. 

R: The full-length M72 antigen genetically fused to ABD was excluded from the assay due to insufficient expression levels

  • The immunogenicity of the ABD domain itself should be discussed. While ABD fusion is intended to enhance delivery, it may also introduce new epitopes or alter antigen processing. The authors should clarify whether ABD alone was tested for immunogenicity or included as a control. 

R: Thank you for your insightful comment. I agree that incorporating additional peptides in the vaccine formulation is not ideal. However, since we stimulate the immune response in vitro using the whole M. bovis antigen (PPDb), the resulting response is specific to those antigens, minimizing the detection of immune reactions against other vaccine components (such as ABD or LPS contaminant). Furthermore, when animals are infected with M. bovis, the induced immune response is primarily driven by the antigens introduced during infection, reducing the likelihood of cross-reactivity to ABD.

  • Endotoxin removal and quantification are not mentioned. Recombinant proteins expressed in  colican be contaminated with endotoxins, which may confound immunological readouts. The authors should report endotoxin levels and describe any removal procedures. 

R: The protein purification procedure was carefully performed to minimize contamination with LPS or endotoxins. However, we cannot entirely exclude the possibility of trace endotoxin levels in the recombinant proteins. Nevertheless, any potential residual contamination did not affect the assay outcomes, as the animals exhibited no signs of exacerbated inflammatory responses indicative of LPS exposure. Importantly, the immune responses observed after infection and in vitro stimulation were specific to M. bovis antigens, since neither the challenge strain nor the antigen used for in vitro stimulation (PPDb—M. bovis does not produce endotoxins) contained LPS

  1. The use of female BALB/c mice, subcutaneous immunization with BCG, and subsequent boosting with M72 or ABDsM72 nanovaccines is appropriate for a preclinical evaluation. The challenge with virulent bovisand assessment of bacterial loads in lungs and spleens are standard endpoints. The experimental design includes appropriate control groups (PBS, BCG alone), sufficient group sizes (n=8), and well-defined immunization and challenge schedules. The use of both pulmonary and systemic bacterial burden as outcomes provides a comprehensive assessment of vaccine efficacy.
  • The choice of BALB/c mice, while common, has inherent limitations for modeling human or bovine TB pathology. Mice do not develop the full spectrum of granulomatous lesions seen in natural hosts, nor do they recapitulate transmission dynamics or chronic/latent infection. The authors should acknowledge these limitations and discuss their implications for translational relevance. 

R: The following text was included in the discussion:An important limitation of this study is that BALB/c mice do not develop granulomatous lesions or recapitulate key aspects of M. bovis infection, such as transmission dynamics and chronic/latent infection. Therefore, evaluation in a more biologically relevant model (e.g., C3HeB/FeJ mice) is critical for a comprehensive assessment of the efficacy of the tested vaccine strategies.”

  • The route of immunization (subcutaneous) and challenge (aerosol) are appropriate, but the rationale for booster timing (2 and 4 weeks post-primary vaccination) should be justified based on immunological kinetics or prior studies.

R: The following sentence was included in the result section: “We selected this vaccination protocol based on its demonstrated efficacy in our prior studies [20,23].”

  • The duration of follow-up post-challenge (4 weeks) may be insufficient to capture long-term protection or memory responses. 

R: We followed the recommendations of Diane J. Ordway and Ian M. Orme, experts in animal models for mycobacterial infections. Their protocols state, 'There may be some bacterial elimination, but by days 30 to 40, the infection reaches a chronic state, with mycobacterial loads stabilizing in the lungs (~10⁴ CFU) and spleen (~10³ CFU)” (doi: 10.1002/0471142735.im1905s30). 

R: The assessment of long-term protection was not within the scope of this study; however, we recognize the importance of evaluating these vaccine strategies over the long term. We included the following sentence in the discussion “In addition, assessing both the long-term protective capacity and the induction of multifunctional T cell responses following vaccination will be essential to fully determine their effectiveness of these vaccines against M. bovis infection.”

  1. Flow cytometry is used to characterize lung-resident CD4+ T cell subsets, focusing on markers associated with protective immunity (CXCR3, KLRG1, PD-1, CCR7). Histopathology assesses granuloma formation and tissue lesions. The immunophenotyping strategy is robust, employing established markers and appropriate gating controls (FMO). The focus on lung-resident T cell subsets is justified by their association with protection in TB models. The use of both functional (CFU reduction) and mechanistic (cellular correlates) endpoints strengthens the study.
  • The selection of markers (CXCR3, KLRG1, PD-1, CCR7) is appropriate, but the lack of functional assays (e.g., cytokine production, proliferation, cytotoxicity) limits the mechanistic interpretation. The authors should consider including or discussing functional readouts to complement phenotypic data. 

R: In previous studies, we observed no correlation between intracellular IFN-γ detection in CD4 T cells and bacterial load. Consequently, we excluded this parameter from the current study. The T cell populations evaluated in this study were defined according to established protocols from previous M. tuberculosis vaccine studies in mice. However, we acknowledge a technical limitation: our analysis did not include physical separation of lung parenchyma from pulmonary vasculature. Instead, we used KLRG1-negative expression as a surrogate marker for lung-homing CD4+ T cells. The lack of evaluation of multifunctional CD4+T cells was included as another limitation of this study. (“. In addition, assessing both the long-term protective capacity and the induction of multifunctional T cell responses following vaccination will be essential to fully determine their effectiveness of these vaccines against M. bovis infection.”).

  • The statistical analysis is generally sound, but the use of multiple comparison corrections and reporting of exact p-values should be clarified. The authors should specify whether data met assumptions for parametric tests and provide effect sizes where possible. 

R: We included effect sizes in the figure legends and specified the statistical test applied for parametric (ANOVA) and non-parametric data (Kruskal-Wallis) .

  • The histopathological scoring system is not fully described. Quantitative criteria for granuloma assessment, lesion grading, and observer blinding should be detailed to ensure objectivity and reproducibility. 

R: The following text was included in M&M: “For microscopic evaluation of lung, liver, and spleen tissues, we assessed histopathological lesions based on the criteria established by Aguilar León et al. (2009) [21]. Granulomas (composed of macrophages and epithelioid cells) were quantified per tissue section, while other features were scored as present or absent: caseous necrosis within granulomas, calcium deposits (mineralization), Langhans giant cells, bronchus-associated lymphoid tissue (BALT), lymphocytic infiltrates, vascular congestion and haemorrhage. Specifically, liver specimens were also evaluated for necrotic foci. All findings were systematically recorded.”

  1. The authors report that both unloaded and antigen-loaded chitosan/alginate nanocapsules exhibit a rounded morphology with a dense oily core, as confirmed by SEM12. DLS measurements indicate hydrodynamic radii of 333–341 nm and zeta potentials ranging from -25 to -38 mV, consistent with stable colloidal suspensions. The detailed characterization of nanocapsule morphology and surface charge supports the integrity and stability of the vaccine formulations. The use of supplementary figures and tables enhances data transparency.
  • The absence of antigen loading efficiency and release kinetics data, as noted above, is a significant gap. These parameters are essential for correlating nanocapsule properties with immunogenicity and efficacy.

R: We have now included quantitative data on protein loading efficiency in the nanocapsules. While we did not perform in vitro antigen release assays, as this study specifically focused on evaluating short-term protective efficacy in only one point time post vaccination, we acknowledge that characterization of antigen release kinetics would be essential for assessing long-term vaccine performance.

  1. SDS-PAGE analysis confirms the expected molecular weights of M72 (~70 kDa) and ABDsM72 (~40 kDa), with successful purification from coli The use of His-tag affinity purification is standard and effective. The clear demonstration of protein expression and purity supports the validity of subsequent immunization experiments.
  • The potential for residual host cell proteins or contaminants should be addressed, particularly regarding immunogenicity and safety. 

R: We agree that contamination with endotoxins such as LPS should be avoided in vaccine formulations. However, since we stimulated immune responses in vitro using whole M. bovis antigen (PPDb), the observed responses were specific to these antigens, minimizing detection of immune reactions against potential vaccine contaminants. Importantly, no inflammatory responses were observed in mice following antigen administration, addressing potential safety concerns. The following sentence was included in M&M section: “The protein purification procedure was carefully performed to minimize contamination with LPS or endotoxins.”

  1. All vaccinated groups (BCG, BCG+M72, BCG+ABDsM72) exhibit significantly reduced lung CFU counts compared to PBS controls, with no significant differences among vaccinated groups. However, only the BCG+M72 group shows a significant reduction in splenic CFUs, suggesting enhanced control of systemic dissemination. The demonstration of significant protection against pulmonary infection by all vaccine regimens is robust and consistent with prior studies. The additional reduction in splenic dissemination by the M72 boost is a notable finding.
  • The lack of enhanced lung protection by booster regimens compared to BCG alone is a key limitation, consistent with other studies in nonhuman primates and mice. The authors should discuss possible reasons, such as the ceiling effect of BCG in murine models, suboptimal antigen dosing, or insufficient adjuvant potency.

R: We included the following paragraph in the discussion: “Several factors may explain the lack of efficacy of these booster vaccines in controlling M. bovis replication in the lungs, including the potentially low adjuvant capacity of the delivery system, the antigen dose administered, or simply a ceiling effect imposed by prior BCG vaccination in the murine model.”

  • The absence of long-term follow-up limits conclusions about durability of protection. The authors should consider or discuss extended challenge studies.

R: We agree with this comment on the discussion of the manuscript and we included the following sentence in this section “In addition, assessing both the long-term protective capacity and the induction of multifunctional T cell responses following vaccination will be essential to fully determine their effectiveness of these vaccines against M. bovis infection. ”

  • The use of only female mice may limit generalizability, as sex differences in TB immunity are documented. The authors should acknowledge this limitation.

R: This text was added in the discussion: “Also, given the reported sex-based differences in tuberculosis vaccine responses in murine models [36,37], evaluating these vaccine candidates in male subjects is also necessary to ensure a complete understanding of their protective potential."

  1. Flow cytometry reveals that BCG+ABDsM72 vaccination significantly increases the frequency of lung CD4+CXCR3+KLRG1- T cells, a subset associated with protective immunity2. PD-1 expression and CCR7+ central memory T cells do not differ significantly among groups. Correlation analysis shows a negative association between lung CFUs and CD4+CXCR3+KLRG1- T cells, but a positive association with KLRG1+ and PD-1+ subsets. The identification of lung-homing, parenchyma-resident CD4+ T cell subsets as correlates of protection is a valuable mechanistic insight. The use of regression analysis to link immune phenotypes with bacterial burden strengthens the conclusions.
  • The lack of functional assays (e.g., cytokine production, recall responses) limits the mechanistic depth. The authors should consider including ELISPOT, intracellular cytokine staining, or proliferation assays to assess T cell functionality. 

R: In previous studies, we observed no correlation between intracellular IFN-γ detection in CD4 T cells and bacterial load. Consequently, we excluded this parameter from the current study. The T cell populations evaluated in this study were defined according to established protocols from previous M. tuberculosis vaccine studies in mice. However, we acknowledge a technical limitation: our analysis did not include physical separation of lung parenchyma from pulmonary vasculature. Instead, we used KLRG1-negative expression as a surrogate marker for lung-homing CD4+ T cells. The lack of evaluation of multifunctional CD4+T cells was included as another limitation of this study. (“. Furthermore, determining both the long-term protective capacity and multifunctional T cell responses following vaccination is essential to fully evaluate their effectiveness against M. bovis infection”).

  • The unexpected positive correlation between PD-1+ T cells and bacterial burden warrants deeper discussion. The dual role of PD-1 in TB protective versus exhaustion should be explored in light of recent literature.

R: To address this reviewer's concern we include the following text in the discussion: “The dual role of PD-1 in immune regulation has gained increasing attention, particularly in the context of oncologic therapies and infectious diseases such as TB. In TB patients, elevated PD-1 expression has been observed on various immune cells, including CD4+ T cells, NK cells, neutrophils, and monocytes, suggesting a role in immune modulation [38]. However, despite its therapeutic value in cancer, anti-PD-1 therapy may disrupt the homeostasis of M. tuberculosis-specific T cells. This disruption can lead to extracellular matrix degradation, enhanced recruitment of monocytes and neutrophils, and dysregulation of key cytokines such as TNF-α— conditions that can potentially favour M. tuberculosis growth [39].

Studies using PD-1-deficient mice and observations from cancer patients undergoing PD-1 blockade underscore the non-redundant role of PD-1 in TB control. Paradoxically, high PD-1 expression on circulating T cells is a well-recognized hallmark of active TB disease [40,41], implying a context-dependent function. Our current findings support this duality: we observed a positive correlation between PD-1+CD4+ T cells and higher bacterial loads in patients with active TB [40]. This positive association suggests that PD-1 upregulation may reflect a host response to active infection rather than a protective immune mechanism induced by vaccination..”

  • The absence of significant changes in CCR7+ central memory T cells should be discussed in the context of vaccine-induced memory and tissue residency.

R: To address this reviewer's concern we include the following text in the discussion: “Central memory T cells (TCM) are phenotypically distinct from effector memory T cells (TEM) due to their expression of CCR7, a chemokine receptor that facilitates migration to secondary lymphoid organs via interactions with CCL19 and CCL21 [38]. In this study, we detected no significant differences in the frequencies of CCR7+ T cells in the lungs across experimental groups. In contrast, our earlier research demonstrated a higher proportion of CD4+CCR7+ T cells in the lungs of vaccinated mice compared to unvaccinated controls, following M. bovis challenge one year post-vaccination. These collective findings suggest that the expansion of CCR7+ T cells may depend on the interval of vaccination and pathogen exposure.

The dual role of PD-1 in immune regulation has gained increasing attention, particularly in the context of oncologic therapies and infectious diseases such as TB. In TB patients, elevated PD-1 expression has been observed on various immune cells, including CD4+ T cells, NK cells, neutrophils, and monocytes, suggesting a role in immune modulation [39]. However, despite its therapeutic value in cancer, anti-PD-1 therapy may disrupt the homeostasis of M. tuberculosis-specific T cells. This disruption can lead to extracellular matrix degradation, enhanced recruitment of monocytes and neutrophils, and dysregulation of key cytokines such as TNF-α— conditions that can potentially favour M. tuberculosis growth [40].”

  1. No significant differences are observed among groups in granuloma formation, macrophage infiltration, or lesion presence. This finding is consistent with the limitations of murine models for recapitulating human or bovine TB pathology. The inclusion of histopathological analysis provides a complementary endpoint to bacterial burden and immune phenotyping.
  • The lack of detailed quantitative scoring and representative images limits interpretability. The authors should provide more granular data and discuss the limitations of murine pathology for modeling bTB.

R: We have included a new supplementary figure displaying representative histopathological images (Fig S4). Additionally, the limitations of the murine model for bovine tuberculosis have been addressed in the manuscript.

  1. The study demonstrates that chitosan nanocapsule-formulated M72 and ABDsM72 antigens, when used to boost BCG vaccination, do not enhance pulmonary protection against bovisin mice but do reduce systemic dissemination (splenic CFUs) and modulate lung-resident T cell populations. The increase in CD4+CXCR3+KLRG1- T cells in the BCG+ABDsM72 group supports the potential of albumin-hitchhiking strategies for improving lymphatic targeting and immune priming. The authors provide a balanced discussion, acknowledging the limitations of their findings and situating them within the broader context of TB and bTB vaccine development. The recognition that BCG’s robust protection in mice may mask incremental improvements from booster regimens is insightful.
  • The discussion could be strengthened by a more critical appraisal of the translational relevance of murine findings. The limitations of mouse models for bTB, including differences in granuloma structure, chronicity, and immune responses, should be explicitly addressed. 

R: As mentioned above we included the following sentences to address this concern “ An important limitation of this study is that BALB/c mice do not develop granulomatous lesions or recapitulate key aspects of M. bovis infection, such as transmission dynamics and chronic/latent infection. Therefore, evaluation in a more biologically relevant model (e.g., C3HeB/FeJ mice) is critical for a comprehensive assessment of the efficacy of the tested vaccine strategies. In addition, assessing both the long-term protective capacity and the induction of multifunctional T cell responses following vaccination will be essential to fully determine their effectiveness of these vaccines against M. bovis infection. Also, given the reported sex-based differences in tuberculosis vaccine responses in murine models [36,37], evaluating these vaccine candidates in male subjects is also necessary to ensure a complete understanding of their protective potential.”

  • The potential implications for DIVA compatibility and field deployment in cattle are mentioned but not fully developed. The authors should elaborate on how their vaccine strategy could be integrated with current diagnostic and control programs. 

R: To address this reviewer's concern, we have added the following sentence to the discussion: “Importantly, the vaccine formulations tested in this study are compatible with a DIVA diagnostic approach, as they rely on the immunodominant antigens ESAT-6, CFP-10, and Rv3615c—proteins not expressed by BCG. This key feature makes the BCG/M72 and BCG/ABDsM72 vaccines promising candidates for field evaluation in cattle.”

  • The mechanistic basis for the observed reduction in splenic dissemination by M72 boosting, but not ABDsM72, should be explored. Possible explanations include differences in antigen processing, trafficking, or immune activation.

R: The following  text was included in the discussion to address this concern. “In contrast, boosting with ABDsM72 did not enhance control of splenic mycobacterial loads. This discrepancy may be due to the absence of the carboxyl-terminal region in ABDsM72, which in M72 may contain critical antigenic peptides necessary for an effective immune response..”.

  • The lack of significant differences in histopathology and memory T cell responses should be discussed in the context of vaccine-induced immunity and protection.

R: We added this paragraph in the results section: “One possible explanation for the lack of histopathology differences between vaccinated and unvaccinated mice is that the analysis was performed during the early stage of granuloma formation, a phase in which correlations with mycobacterial loads in organs are not typically observed. In fact, our previous studies found no correlation between the number of macrophage-rich granulomas and CFU counts in the mouse lungs [24]..”

  1. The findings are consistent with prior studies showing that BCG prime–subunit boost regimens (including M72/AS01E) often fail to enhance pulmonary protection in animal models, despite inducing robust immune responses. The partial efficacy of M72/AS01E in human trials (approximately 50% protection) underscores the challenge of translating preclinical findings to the clinic. The authors appropriately reference recent trials and reviews, situating their work within the evolving landscape of TB vaccine research.
  • A more detailed comparison with other adjuvant and delivery strategies (e.g., viral vectors, alternative nanocarriers) would enhance the discussion. The potential advantages and disadvantages of chitosan nanocapsules relative to other platforms should be articulated. 

R: We acknowledge the importance of including comparisons with other adjuvants, particularly those studies that evaluated the same antigen with different adjuvant formulations. For this purpose, we have added the following paragraph to the Discussion:"Earlier formulations of M72 conferred protection in mice and guinea pigs, although boosting BCG with M72/AS02A did not reduce mouse lung bacterial loads compared to BCG alone [33]. Skeiky and collaborators found that M72 formulated in AS01B elicited a stronger CD8+ T cell response (CTL and IFN-γ) than M72 formulated with AS02 in immunized C57BL/6 mice [23], highlighting the critical role of adjuvant selection in vaccine efficacy. 

In the present study, we conjugated recombinant antigens to chitosan/alginate nanocapsules, a decision based on our prior findings demonstrating that chitosan nanocapsules loaded with the fusion protein H65 enhanced protection against M. bovis challenge in mice, as indicated by reduced bacillary load in lung tissue [23]. "

  1. The authors suggest that longer-term studies, alternative animal models (e.g., guinea pigs, cattle), and mechanistic investigations are warranted. These recommendations are appropriate and align with the study’s limitations.
  • The authors should outline specific experimental plans for addressing key gaps, such as testing in natural hosts, evaluating mucosal delivery routes, or optimizing antigen/adjuvant combinations.

R: To address this reviewer's concern, we have added the following paragraphs to the discussion: “Importantly, the vaccine formulations tested in this study are compatible with a DIVA diagnostic approach, as they rely on the immunodominant antigens ESAT-6, CFP-10, and Rv3615c—proteins not expressed by BCG. This key feature makes the BCG/M72 and BCG/ABDsM72 vaccines promising candidates for field evaluation in cattle.”

“Future studies should explore mucosal delivery routes and optimize antigen-nanocapsule formulations to enhance the efficacy of the vaccine candidates beyond the current findings.” 

  • The potential for chitosan nanocapsules to enhance mucosal immunity, given their mucoadhesive properties, should be discussed in greater detail.

R: The following text was included at the end of the discussion: “Future studies should explore mucosal delivery routes and optimize antigen-nanocapsule formulations to enhance the vaccine candidates' performance beyond the current findings.”

  1. The study employs appropriate controls, randomization, and blinding where applicable. The statistical analyses are generally sound, though additional details on data distribution, effect sizes, and corrections for multiple comparisons are needed. The use of multiple complementary endpoints (CFU counts, flow cytometry, histopathology) enhances internal validity. The consistency of findings with prior studies supports the reliability of the results.
  • Potential confounders, such as endotoxin contamination, batch variability in nanocapsule preparation, and sex-specific effects, are not fully addressed. The authors should report measures taken to control for these variables. 

R: As previously noted, endotoxin contamination was undetectable in mice, as evidenced by the absence of exacerbated inflammatory responses. The vaccines were formulated using a single batch of nanocapsules, ensuring consistency across all preparations. Additionally, all animals in this study were female, eliminating potential sex-specific effects. These potential confounders were addressed in the manuscript. 

  1. The generalizability of findings to natural hosts (cattle), other vaccine platforms, and field conditions is limited by the use of a murine model and controlled laboratory settings. The authors appropriately acknowledge these limitations.
  • The authors should discuss the steps needed to translate their findings to cattle or other relevant species, including dose scaling, safety assessment, and regulatory considerations. 

R: We agree on the need to define the regulatory pathway our vaccine candidate should follow to be approved and adopted for animal tuberculosis control. However, we consider the current results preliminary, and as mentioned in the manuscript, further testing of additional variables is required to enhance the protective efficacy of our vaccine candidates before reaching that stage

  • The potential for chitosan-based vaccines to interfere with or complement existing diagnostic tests (e.g., DIVA assays) should be elaborated. 

R: Since BCG lacks expression of key DIVA antigens (ESAT-6, CFP-10, and Rv3615c), the use of either BCG/M72 or BCG/ABDsM72 vaccine candidates in cattle poses no risk of interference with DIVA diagnostic testing. The following text was included in the discussion: “Importantly, the vaccine formulations tested in this study are compatible with a DIVA diagnostic approach based on the immunodominant antigens ESAT-6, CFP-10, and Rv3615c, as these proteins are not expressed by BCG. This remarkable feature makes the BCG/M72 and BCG/ABDsM72 vaccines promising candidates to move to field evaluation in cattle.”